# Solid Fe Resources Separated from Rolling Oil Sludge for CO Oxidation

**DOI:** 10.3390/ijms232012134

**Published:** 2022-10-12

**Authors:** Wei Gao, Sai Tang, Ting Wu, Jianhong Wu, Kai Cheng, Minggui Xia

**Affiliations:** Hubei Key Laboratory of Biomass Fibers and Eco-Dyeing & Finishing, School of Chemistry and Chemical Engineering, Wuhan Textile University, Wuhan 430200, China

**Keywords:** ROS, solid Fe resources, recycling, CO oxidation

## Abstract

The efficient recycling of valuable resources from rolling oil sludge (ROS) to gain new uses remains a formidable challenge. In this study, we reported the recycling of solid Fe resources from ROS by a catalytic hydrogenation technique and its catalytic performance for CO oxidation. The solid Fe resources, after calcination in air (Fe_2_O_3_-H), exhibited comparable activity to those prepared by the calcinations of ferric nitrate (Fe_2_O_3_-C), suggesting that the solid resources have excellent recycling value when used as raw materials for CO oxidation catalyst preparation. Further studies to improve the catalytic performance by supporting the materials on high surface area 13X zeolite and by pretreating the materials with CO atmosphere, showed that the CO pretreatment greatly improved the CO oxidation activity and the best activity was achieved on the 20 wt.%Fe_2_O_3_-H/13X sample with complete CO conversion at 250 °C. CO pretreatment could produce more oxygen vacancies, facilitating O_2_ activation, and thus accelerate the CO oxidation reaction rate. The excellent reducibility and sufficient O_2_ adsorption amount were also favorable for its performance. The recycling of solid Fe resources from ROS is quite promising for CO oxidation applications.

## 1. Introduction

Rolling oil is extensively used for refrigeration and lubrication during the cool rolling process of stainless steel, but it will deteriorate after long-term use and transform into rolling oil sludge (ROS) by mixing with iron powder. ROS is one type of refractory hazardous waste [1,2,3]. With the rapid development of the steel industry, the enormous usage of rolling oil produces a tremendous increase of ROS, causing serious threats to both the ecological environment and human health [3,4,5]. Conventional treatment technologies to dispose of ROS include incineration, distillation, brick or briquette making, solvent extraction, etc. [6,7,8], but encounter more problems in practical use, such as high operation cost, complex process, serious waste pollution, insufficient use of calorific value, etc. [8,9]. Hydrofining technology is one of the most common technologies for the recycling of spent lubrication oil [10,11,12]. It possesses many advantages, such as a large treatment capacity, high recovery rate, good product quality, etc., despite its huge initial input costs. For ROS treatment, the distillation process is generally employed to cut components prior to the hydrofining process, and solid wastes are not recycled effectively [13,14,15]. Moreover, the solid wastes are also hard to separate from ROS even after the hydrofining treatment due to the addition of solid catalysts, which could mix with the solid wastes and be hardly separated. The Fe element does not have full electron fill in d-shell with a body-centered cubic lattice structure, thereby exhibiting a certain hydrogenation capability [16,17]. Hence, the development of a simple technology that can effectively recycle the high-value solid resources from ROS and thereby find the potential application of these solid resources is of great importance.

Catalytic CO oxidation is a well-known reaction in heterogeneous catalysis. It can not only act as a prototypical reaction to understand the reaction mechanism but also receive wide applications in practice, such as CO abatement in closed systems (e.g., submarines, aircraft, spacecraft, etc.), automotive exhaust catalytic purification, gas purification in gas masks, removal of CO impurity in feed gas, etc. [18,19,20]. It has been reported that the pretreatment of catalysts with the reactant is of great importance and can significantly affect the catalytic performance for CO oxidation. For example, the pretreatment of Pd-supported catalysts with H_2_ or CO showed excellent catalytic performance due to the formation of new active sites [21] and the improved ability for O_2_ activation [22]. Fe-based catalysts are extensively studied as promising alternatives to the commercial noble metal-based three-way catalysts, such as Fe-Ce [23], Fe-Co [24], Fe-Ni [25], etc. Tian et al. [26] synthesized Cu-Fe-Co ternary oxide thin film supported on copper grid mesh for CO oxidation, showing excellent catalytic activity, which was owed to the synergistic effects of chemisorbed oxygen species, electrical resistivity, easy mass transferability, etc. Moreover, 13X zeolite is usually employed as catalyst support due to its high specific surface area, easy availability, and low cost. It has been widely applied in various reactions, including catalytic removal of ethylbenzene, cyclohexane, and hexadecane [27,28]. Therefore, it is quite interesting to develop Fe-supported 13X zeolite catalysts for CO oxidation.

In this work, ROS collected from a food-grade stainless steel facility in Wuhan Iron and Steel Co. (WISCO) was disposed of by the hydrogenation technique. The solid Fe resources can act as catalysts for the hydrogenation of organics in the ROS and be automatically separated from the oil without adding extra hydrogenation catalysts. To explore the application of the recycled solid Fe resources from the ROS, they were utilized as raw materials to prepare catalysts by supporting them on 13X zeolite. The catalytic performance of such prepared catalysts was evaluated for CO oxidation reaction at a stoichiometric feed and the effect of CO pretreatment on the catalytic activity was investigated.

## 2. Results and Discussion

### 2.1. Physical Properties and Morphology of Fe Based Catalysts

Figure 1 shows the images of the ROS and the separated oil and solid phase after the catalytic hydrogenation process. The ROS was seriously emulsified with high viscosity and was difficult to dry naturally, owing to the poor volatile ability of the oily sludge components. Meanwhile, it emitted funky odors. After catalytic hydrogenation, the product was readily filtered to achieve complete oil/solid separation. The oil color changed from black to brown, and the viscosity became quite low with the capability of free flow. The solid was quite loose and easily crushed into powder. It was then used to prepare Fe-based catalysts for CO oxidation.

HRTEM/TEM images in Figure 2 show that the Fe_2_O_3_-H sample (the solid Fe resource after air calcination) was made of nanoparticles in the size range of 30~40 nm. The interplanar distances were measured to be 0.27 and 0.25 nm, which were assigned to the (104) and (111) planes of Fe_2_O_3_, respectively. To detect the crystalline phases of the Fe-based catalysts, XRD measurements were performed and the results are presented in Figure 3a. By screening the XRD database, the peaks located at 24.1, 33.2, 35.6, 40.8, and 49.6° are assigned to those of the α-Fe_2_O_3_ phase (PDF #73-2234), [29,30], which is consistent with the TEM/HRTEM results. For the 13X zeolite-supported catalysts, the catalysts displayed the characteristic diffraction peaks ascribed to 13X zeolite besides α-Fe_2_O_3_ phase. Meanwhile, the intensity of diffraction peaks of 13X zeolite weakened in accordance with that reported elsewhere [27]. Figure 3b depicts the XRD patterns of samples pretreated with CO. For the 13X zeolite-supported samples (20% Fe_2_O_3_-C/13X@CO and 20% Fe_2_O_3_-H/13X@CO), the new diffraction peaks at 30.3, 37.4, 43.4, 53.9, and 57.5° were attributed to Fe_3_O_4_ (PDF#88-0315) indicated that the Fe_2_O_3_ in these two samples was reduced to Fe_3_O_4_ by CO. While for the unsupported samples (Fe_2_O_3_-C@CO and Fe_2_O_3_-H@CO), the peaks ascribed to Fe_2_O_3_ co-existed with those ascribed to Fe_3_O_4_. These results showed that a portion of Fe^3+^ in the fresh samples was converted into Fe^2+^ after exposure to 1% CO/N_2_ at 450 °C.

### 2.2. Redox and Surface Properties

To investigate the redox properties of the catalysts, H_2_-TPR experiments were carried out. As shown in Figure 4a, the reduction behavior of iron species was divided into three stages for the Fe_2_O_3_-H sample: Fe_2_O_3_ to Fe_3_O_4_ in the temperature range of 300–450 °C, and Fe_3_O_4_ to FeO, then to Fe^0^ in the high temperature range of 450–800 °C [31,32,33]. The introduction of 13X zeolite substantially changed the character of the reduction curves. Only one reduction peak at approximately 533 °C was observed for the 20% Fe_2_O_3_-H/13X sample. The reduction of Fe_2_O_3_-H/13X was almost completed at 700 °C, which was much lower than that of the Fe_2_O_3_-H sample, indicating that the presence of 13X zeolite facilitated the reduction of iron species. In contrast, the reduction was quite complex for 20% Fe_2_O_3_-C/13X. It started off at a similar temperature to that of the 20% Fe_2_O_3_-H/13X, but the profile then became flat until a major peak appeared at 792 °C. The poor reducibility of 20% Fe_2_O_3_-C/13X could account for the low activity as discussed below. The CO pretreatment affected the redox ability of the catalysts, and the corresponding results are shown in Figure 4b. Fe species of Fe_2_O_3_-H@CO were readily reduced as the reduction process was completed at lower temperatures compared to Fe_2_O_3_-H. Notably, CO treatment led to the increment of the reduction peak area of the Fe_2_O_3_-H and 20% Fe_2_O_3_-H/13X samples compared to their respective fresh samples, but the opposite trend was observed on the Fe_2_O_3_-C-based catalysts. The results indicated that CO pretreatment facilitated the generation of reducibly Fe active species on the Fe_2_O_3_-H-based samples while inhibiting it on the Fe_2_O_3_-C-based samples. Moreover, the reduction peak of 20% Fe_2_O_3_-C/13X after CO pretreatment remained at around 790 °C, while that of 20% Fe_2_O_3_-H/13X shifted from 533 to 515 °C after CO exposure, showing that the redox ability was improved on the 20% Fe_2_O_3_-H/13X@CO.

To investigate the O_2_ adsorption behavior of samples, O_2_-TPD experiments of fresh samples were conducted, and the results are shown in Figure 5a. The desorption peaks below 300 °C and above 600 °C are generally attributed to chemisorbed oxygen species and bulk lattice oxygen, respectively [34,35]. Both Fe_2_O_3_-C and Fe_2_O_3_-H showed a weak ability for O_2_ adsorption, while the introduction of 13X zeolite profoundly enhanced the O_2_ adsorption ability, which was a crucial factor influencing the activity of CO oxidation. Compared to 20% Fe_2_O_3_-C/13X, 20% Fe_2_O_3_-H/13X exhibited a bigger O_2_ desorption peak, showing its superior adsorption capability. The effect of CO pretreatment on O_2_ adsorption ability was investigated, and the results are shown in Figure 5b. Lower adsorption temperatures were observed on all the samples after CO pretreatment. Similarly, the 13X zeolite-supported Fe_2_O_3_ samples showed a larger O_2_ adsorption amount than pure Fe_2_O_3_ samples after CO pretreatment.

An XPS measurement was employed to investigate the chemical state of the samples. XPS spectra of Fe 2p of samples before and after CO pretreatment are presented in Figure 6. The peaks at 710.97 and 724.45 eV are assigned to Fe 2p_3/2_ and Fe 2p_1/2_, respectively, and the satellite peak at 719.16 eV is attributed to Fe_2_O_3_ [36,37]. No peaks assigned to Fe^0^ and Fe^2+^ were detected, as expected. The binding energies of Fe 2p shifted to lower values at 0.5 eV when Fe_2_O_3_-H and Fe_2_O_3_-C were supported on 13X zeolite. This indicated that the introduction of 13X zeolite affected the chemical environments of Fe^3+^ in the samples. Three peaks at 709.6–709.9 eV, 710.2–711.7 eV, and 712.9–713.7 eV appeared after CO pretreatment, which can be assigned to octahedral Fe(Ⅱ) species, octahedral Fe(Ⅲ) species, and tetrahedral Fe(Ⅲ) species [38]. This suggests that the iron in the samples existed both in Fe^2+^ and Fe^3+^ species. Furthermore, a shakeup satellite attributed to the Fe 2p_3/2_ peak appeared, which was ~9 eV higher than that of the main Fe 2p_3/2_ peak. Table 1 lists the oxidation states of Fe species in the samples after CO pretreatment. The high Fe^2+^ / (Fe^2+^ + Fe^3+^) ratio caused the imperfect structure of iron oxide and resulted in the formation of oxygen vacancies that facilitated CO oxidation. The lowest Fe^2+^ content was found in 20% Fe_2_O_3_-C/13X@CO, and Fe_2_O_3_-H@CO exhibited a higher Fe^2+^ / (Fe^2+^ + Fe^3+^) ratio than Fe_2_O_3_-C@CO. The Fe^2+^ / (Fe^2+^ + Fe^3+^) ratio increased from 0.296 for Fe_2_O_3_-H@CO to 0.346 for 20% Fe_2_O_3_-H/13X@CO, which can be attributed to the interaction between iron and 13X zeolite.

Figure 7 shows the XPS spectra of O 1s of samples before and after CO pretreatment, which could be fitted into three peaks assigned to chemisorbed oxygen O_β_ species (O_2_^2−^, ~531.3 eV; and O_2_^−^, ~533.0 eV) and lattice oxygen O_α_ species (O^2-^, ~529.8 eV), respectively. Table 1 lists the O_α_ and O_β_ ratio calculated from the fitted peak areas ascribed to O_α_ and O_β_ species. The O_β_ ratio increased from 28.8 to 33.6% on Fe_2_O_3_-H catalysts when compared to Fe_2_O_3_-C catalysts. As lattice oxygen O_α_ species were closely related to iron oxides in the samples, the introduction of 13X zeolite changed the major feature of the profile, showing that O_β_ ratio was much higher than O_α_ ratio due to low Fe contents. CO pretreatment also markedly promoted the generation of chemisorbed oxygen O_β_ species (O_2_^2−^, ~531.3 eV; and O_2_^−^, ~533.0 eV), e.g., the O_β_ ratio increased from 82.0% to 88.5% after the 20% Fe_2_O_3_-H/13X was pretreated with CO. The results were consistent with the Fe 2p XPS spectra, i.e., more Fe species with low valence states were favorable for the generation of more oxygen vacancies. O_β_ species are widely recognized as more reactive than O_α_ species because of their higher mobility.

### 2.3. CO Oxidation Application

#### 2.3.1. Effect of Calcination Temperatures, Different Synthesis Methods, and Pretreatment Conditions

The phase of Fe_2_O_3_-H, determined by calcination temperatures, showed a significant influence on CO oxidation performance. Appendix A presents XRD patterns of Fe_2_O_3_-H under different calcination temperatures. Several main characteristic diffraction peaks at 30.2, 35.6, 43.3, 53.7, and 57.3° appeared on the samples under 400 and 500 °C calcination, which are assigned to metastable maghemite phase *γ*-Fe_2_O_3_ (PDF#39-1346). Further increasing the calcination temperatures to 600 and 700 °C led to the phase transformation to the more stable hematite phase *α*-Fe_2_O_3_ (PDF#33-0664), which displayed main characteristic diffraction peaks at 24.1, 33.2, 35.6, 40.8, and 49.6°. The average crystalline size, estimated by the Scherrer equation, slightly increased from 25.1 to 27.9 nm when increasing the calcination temperatures from 400 to 600 °C, while it markedly increased to 35.8 nm when further elevating the calcination temperature to 700 °C. This suggests that the crystalline size would significantly grow large at calcination temperatures above 600 °C. Appendix A exhibited CO oxidation activity over Fe_2_O_3_-H after calcined at 400–700 °C. The CO oxidation performance increased in the following sequence of 400 < 500 < 700 < 600 °C, suggesting that the optimal calcination temperature was 600 °C.

The activities of the original solid Fe resources, Fe_2_O_3_-H and Fe_2_O_3_-C, were then compared, and the corresponding results are shown in Figure 8. The solid Fe resources showed the lowest CO conversion in the whole temperature window compared with the samples after calcination, which may be due to residual oil or other impurities in the solid Fe resources that inhibited its catalytic activity. Fe_2_O_3_-H and Fe_2_O_3_-C showed similar activity with T10 at 200 °C and T100 at 300 °C, demonstrating that the performance of Fe_2_O_3_-H, obtained from the ROS, was comparable to that of the synthetic Fe_2_O_3_ in the lab, and hence the excellent recycling value of the solid Fe resources in CO oxidation application. CO pretreatment could change the chemical state of active species and catalyst structure, thus further affecting CO oxidation activity. Figure 8b shows CO oxidation performance over Fe_2_O_3_-H, Fe_2_O_3_-H@CO, and Fe_2_O_3_-C@CO. It was found that the activity of Fe_2_O_3_-H and Fe_2_O_3_-C can be significantly improved after CO pretreatment, and the Fe_2_O_3_-H@CO exhibited slightly higher activity than the Fe_2_O_3_-C@CO. This indicates that CO pretreatment can evidently boost the activity of Fe_2_O_3_ for CO oxidation. The best activity was obtained from Fe_2_O_3_-H@CO, which showed 54% and 100% CO conversions at 200 and 250 °C, respectively. According to XRD and XPS results, a portion of Fe^3+^ was reduced to low valence Fe^2+^ species during the CO pretreatment, resulting in the formation of more oxygen vacancies that facilitated O_2_ adsorption and activation. Enhanced O_2_ adsorption and activation would accelerate CO oxidation rates on the Fe_2_O_3_-H and Fe_2_O_3_-C after CO pretreatment.

#### 2.3.2. CO Oxidation Activity on 13X Zeolite-Supported Catalysts

Pure nanometer oxides readily agglomerate and grow when exposed to heat. High surface area supports are typically employed to disperse those nanometer oxide particles to enhance their durability and thermal stability. On this basis, we prepared Fe_2_O_3_-H supported on the 13X zeolite, 5A, FCC and *γ*-Al_2_O_3_ catalysts, and their XRD patterns are shown in Appendix A. All the samples displayed the characteristic peaks ascribed to the corresponding supports, indicating that each support was successfully introduced. Their CO oxidation activities were further compared, and the results are shown in Appendix A. The CO oxidation activity increased in the following order of Fe_2_O_3_-H/FCC < Fe_2_O_3_-H/5A < Fe_2_O_3_-H/*γ*-Al_2_O_3_ < Fe_2_O_3_-H/13X. To further clarify the effect of pure supports on the performance, we tested their catalytic activities. All the supports showed quite poor activities in the whole temperature window. The results suggested that the activity came from Fe_2_O_3_ and the interactions between Fe_2_O_3_ and the support, and that the optimal catalyst support was 13X zeolite.

After selecting 13X zeolite to be a support, the effect of different Fe_2_O_3_-H loadings on the activity was investigated. Figure 9a shows that the CO oxidation activity increased with the increase of Fe_2_O_3_-H loading, and the pure Fe_2_O_3_-H exhibited the best performance. After CO pretreatment, the activity of samples for CO oxidation was markedly enhanced, and the best activity was observed on 20% Fe_2_O_3_-H/13X@CO in Figure 9b. In particular, the 20% Fe_2_O_3_-H/13X@CO showed 71% CO conversion at 200 °C, compared with 5% CO conversion on the 20% Fe_2_O_3_-H/13X at the same temperature, revealing the importance of CO pretreatment in improving the activity of catalysts. CO pretreatment was also introduced to the 20% Fe_2_O_3_-C/13X for comparison. The CO conversion of 20% Fe_2_O_3_-C/13X@CO was 15% at 200 °C, which is much lower than that of 20% Fe_2_O_3_-H/13X@CO. This clearly demonstrated the value of recycling the solid Fe resources from the ROS. The 20% Fe_2_O_3_-H/13X@CO exhibited the highest CO oxidation activity due to better redox ability and superior O_2_ adsorption and activation ability after CO treatment. It had T100 (the catalyst temperature required to reach 100% CO conversion) at 250 °C, which was obviously improved compared with these Fe_2_O_3_-based catalysts reported in the literature, as listed in Table 2. The stability of 20% Fe_2_O_3_-H/13X@CO for CO oxidation was also investigated by evaluating the activities with time-on-stream tests. Appendix A showed that minor CO conversion loss was observed in 30 h, which further proved its great promise for practical applications.

Based on the above results, the catalytic mechanism of CO oxidation on the Fe_2_O_3_-H/13X@CO was then discussed. It is generally believed that iron oxide catalysts follow the redox mechanism in CO oxidation. For supported Fe catalysts, CO molecular first adsorbs on the surface active sites, and is subsequently oxidized by lattice oxygen in iron oxides. Several medium CO_3_^2−^ or HCO_3_^−^ species are formed, which are further converted into CO_2_. Correspondingly, reduced active sites in the former step are, in turn, oxidized to the initial state by adsorbed O_2_ to become available for further reaction cycles. Combined with characterization and activity results, Fe_2_O_3_-H/13X@CO showed the best CO oxidation activity, possibly due to the following reasons: Fe species with low valence states can generate more oxygen vacancies, involving the reaction. Furthermore, the 20% Fe_2_O_3_-H/13X@CO surface provided sufficient chemical chemisorbed oxygen species for reduced active sites’ regeneration and exhibited excellent reducibility for CO molecules’ oxidation, which also contributed to its best CO oxidation performance.

## 3. Methods and Materials

### 3.1. Materials and Reagents

ROS was collected from a food-grade stainless steel facility in WISCO. The facilities used in experiments include a heating mantle, a thermostatic water bath, an electrically-heated drying cabinet, an oven, and a large capacity centrifuge. Raw materials, such as iron(III) nitrate nonahydrate (Fe(NO_3_)_3_·9H_2_O, 99%), ethyl acetate, ammonium hydroxide (NH_3_·H_2_O, 25 wt.%), and 13X zeolite, purchased from Sinopharm Group Chemical Reagent Beijing Co. Ltd., Beijing, China are analytically pure and can be used without further purification.

### 3.2. Catalytic Hydrogenation Reaction

To recycle the solid Fe resources from ROS, catalytic hydrogenation of the ROS was carried out on a 500 mL high-pressure reactor. Typically, 250 g of ROS is added to the reactor. Prior to reaction, the reactor was flushed with nitrogen twice to remove the air residue. Afterwards, H_2_ was filled into the reactor up to 6 MPa at room temperature, and the temperature was raised at a ramp rate of 5 °C/min with a stirring speed of 300 r/min. The reaction was carried out for 4 h at 320 °C to guarantee completion of the hydrogenation process. After cooling to room temperature, the mixture was filtered and the solid Fe resources were finally obtained.

### 3.3. Catalyst Preparation

The solid Fe resources were washed repeatedly with ethyl acetate to remove oil and other impurities, then dried at 100 °C overnight. The powder was calcined at 400–700 °C for 5.5 h, and the resultant product was denoted as Fe_2_O_3_-H. The supported catalysts were synthesized by a mechanical milling method through mixing the solid Fe resources and 13X zeolite. Specifically, the Fe_2_O_3_-H was mixed with 13X zeolite at the desired mass ratio and milled in a rock grinder for 10 min. The Fe_2_O_3_ loading was set at 10~50 wt.% and the catalysts were denoted as x% Fe_2_O_3_-H/13X. The obtained samples were further crushed into 40~60 mesh before use. For comparison, a sample denoted as Fe_2_O_3_-C was obtained through direct calcination of Fe(NO_3_)_3_·9H_2_O at 600 °C for 5.5 h, which was then supported on 13X zeolite (denoted as x%Fe_2_O_3_-C/13X) with the mechanical milling method described above. To avoid confusion, for catalysts pretreated with CO before the reaction, they were denoted as catalyst@CO, e.g., Fe_2_O_3_-H@CO means the fresh Fe_2_O_3_-H was pretreated with 1%CO at 450 °C for 1 h. The scheme of the Fe_2_O_3_-H/13X@CO preparation process was depicted in Figure 10. Moreover, 5A zeolite, fluid catalytic cracking (FCC) spent catalysts, and *γ*-Al_2_O_3_ carriers were chosen to investigate the effect of different supports on CO oxidation activity. The preparation method was the same as Fe_2_O_3_-H/13X except for replacing 13X with 5A, FCC or *γ*-Al_2_O_3_ carriers and Fe_2_O_3_ loading content was fixed at 20%. The catalysts were denoted as Fe_2_O_3_-H/5A, Fe_2_O_3_-H/FCC and Fe_2_O_3_-H/*γ*-Al_2_O_3_.

### 3.4. Catalyst Characterization

X-ray Diffraction (XRD) measurements were performed on a Bruker D8 diffractometer equipped with Cu Kα radiation, with 2θ range between 10° and 80° and a scanning rate of 5°/min. N_2_ sorption isotherms of the catalysts were recorded with a JWGB Sci & Tech Ltd. JW-BK112 analyzer at −196 °C. The samples were degassed at 350 °C for 12 h prior to measurements. A Transmission Electron Microscope (TEM) was conducted on a Tecnai G^2^ F20 S-TWIN to observe surface morphology and measure particle size. X-ray photoelectron spectroscopy (XPS) measurements were performed on a Thermo Escalab 250 Xi spectrometer (Al Kα radiation) to analyze surface chemical composition and chemical states. Binding energies were calibrated using the C1s peak of adventitious carbon at 284.6 eV. A temperature programmed reduction with hydrogen (H_2_-TPR) was performed on a Chemisorb 2720 TPx chemisorption analyzer. A 100 mg sample was pretreated by Ar (25 mL/min) at 350 °C for 30 min and then cooled to room temperature in Ar flow. Afterwards, the TPR spectrum was then collected by raising the temperature at a heating rate of 10 °C/min in 10% H_2_/Ar (50 mL/min). Temperature-programmed desorption of oxygen (O_2_-TPD) was performed on the same apparatus. The first step was to heat up the catalyst to 350 °C in flowing N_2_ to desorb gases adsorbed on the catalyst during catalyst preparation. A 50 mg sample was subjected to pure O_2_ gas at room temperature for 1 h. Afterwards, the gas was switched to pure N_2_ to remove the physically adsorbed O_2_ molecules. The TPD spectrum was then collected by raising the temperature at a heating rate of 10 °C/min.

### 3.5. CO Oxidation Performance Evaluation

CO oxidation performance was evaluated in a home-made fixed bed quartz reactor operating at atmospheric pressure. Typically, 3 g of catalyst was packed in the reactor and supported by a quartz wool plug. To activate the fresh catalysts, the catalysts were heated up to 450 °C for 30 min in flowing N_2_ (100 mL/min). All activities were tested under steady state conditions with a temperature range of 50~450 °C and a temperature increment of 50 °C. The feed gas composition was 0.5 mol.% CO and 0.25 mol.% O_2_ (N_2_ balance) with a stoichiometric feed condition (λ = 1.0). The total flow rate was 100 mL/min and the gaseous hourly space velocity, 2000 h^−1^. CO and CO_2_ concentrations in the effluent were quantified by an on-line gas chromatograph (GC-9560, Zhongke Huifen) equipped with a flame ionization detector (FID). Before entering FID, CO and CO_2_ were fully converted to CH_4_ by a Ni catalyst maintained at 380 °C. The CO conversion was calculated using the following equation:$$X_{CO}\;\left(\%\right)=\frac{C_{CO,in}-C_{CO,out}}{C_{CO,in}}\times 100$$
where $C_{CO,in}$ and $C_{CO_{2},out}$ are the inlet and outlet concentrations of CO, respectively.

## 4. Conclusions

The solid Fe resources, recycled from ROS by the catalytic hydrogenation process, were used as raw materials to prepare Fe-based catalysts for CO oxidation. The XRD and HRTEM results confirmed that the Fe_2_O_3_-H, prepared by calcinating the solid Fe resources in air, mainly consists of the Fe_2_O_3_ phase. TEM results showed that the particle morphology is spherical in nature and the particle size was around 30–40 nm. The CO oxidation activity of Fe_2_O_3_-H was comparable to that of the Fe_2_O_3_-C synthesized in the lab. In particular, the activity of Fe_2_O_3_-H for CO oxidation can be dramatically enhanced after mixing with 13X zeolite and pre-treating in a CO atmosphere, with the best activity obtained from 20% Fe_2_O_3_-H/13X@CO, which showed 71% CO conversion at 200 °C and 100% conversion at 250 °C. This potentiates the great value of recycling the solid Fe resources from ROS. The excellent activity of 20% Fe_2_O_3_-H/13X@CO was attributed to the generation of low-valence Fe species, enhanced reducibility, and improved O_2_ adsorption ability. This work provides a new vista to develop promising alternatives to noble metal-based three-way catalysts via recycling the solid phase of waste ROS.

## Figures and Tables

**Figure 1 ijms-23-12134-f001:**
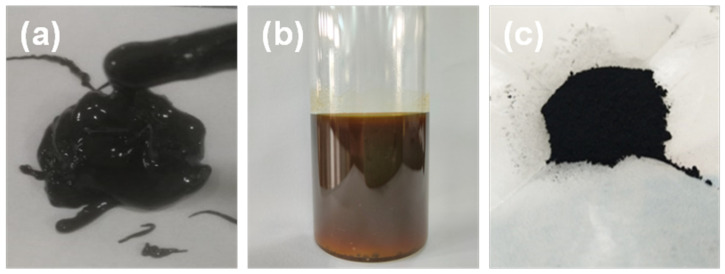
Images of the (**a**) rolling oily sludge, (**b**) separate oil phase and (**c**) solid phase after the catalytic hydrogenation process.

**Figure 2 ijms-23-12134-f002:**
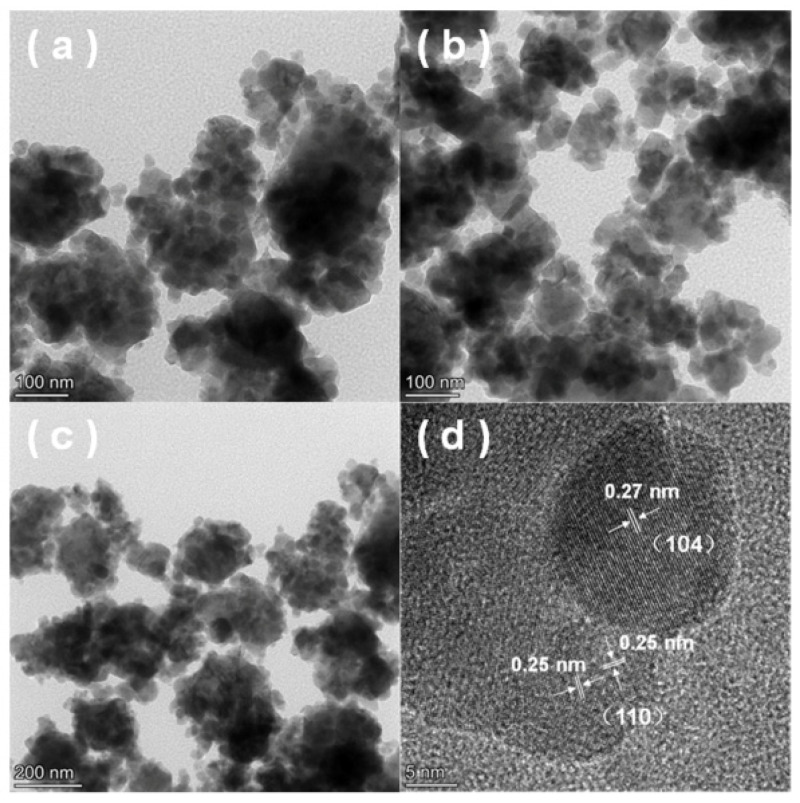
(**a**–**c**) TEM and (**d**) HRTEM images of the Fe_2_O_3_-H.

**Figure 3 ijms-23-12134-f003:**
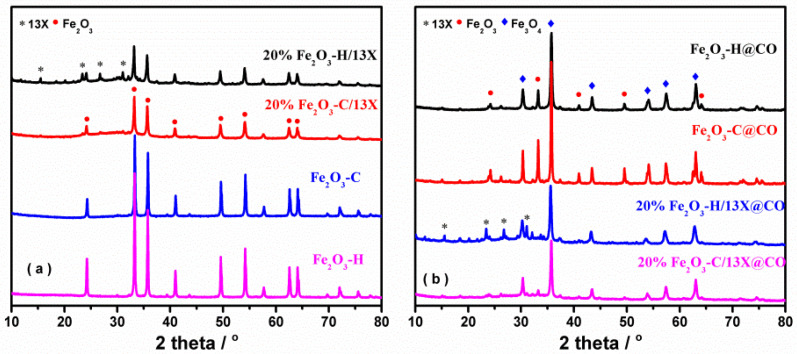
X-ray diffraction patterns on the samples (**a**) without and (**b**) with CO pretreatment.

**Figure 4 ijms-23-12134-f004:**
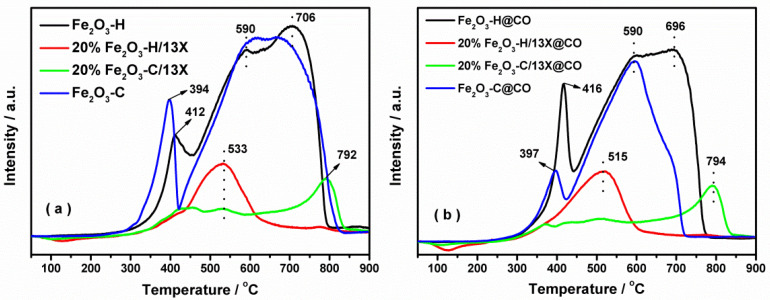
H_2_-TPR profiles on the samples (**a**) without and (**b**) with the exposure to CO.

**Figure 5 ijms-23-12134-f005:**
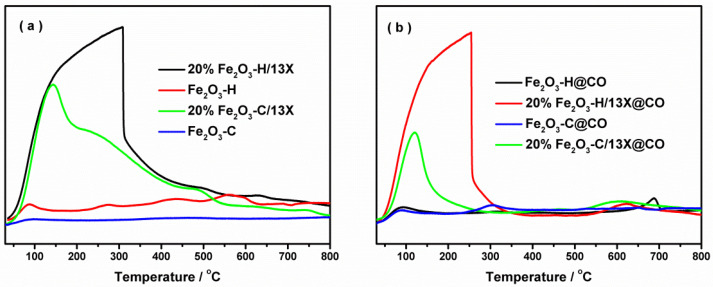
O_2_-TPD profiles on the samples (**a**) without and (**b**) with the exposure to CO.

**Figure 6 ijms-23-12134-f006:**
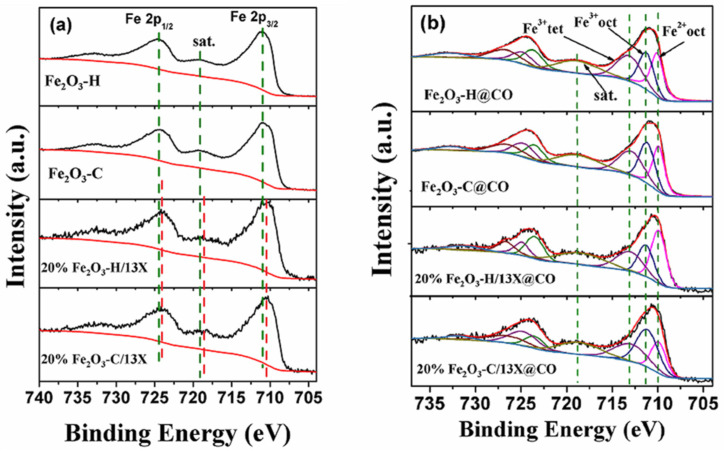
Fe 2p XPS spectra of Fe_2_O_3_-H, Fe_2_O_3_-C, 20% Fe_2_O_3_-H/13X, and 20% Fe_2_O_3_-C/13X (**a**) before and (**b**) after CO pretreatment. Magenta, navy and purple lines represented octahedral Fe(Ⅱ) species, octahedral Fe(Ⅲ) species, and tetrahedral Fe(Ⅲ) species, respectively.

**Figure 7 ijms-23-12134-f007:**
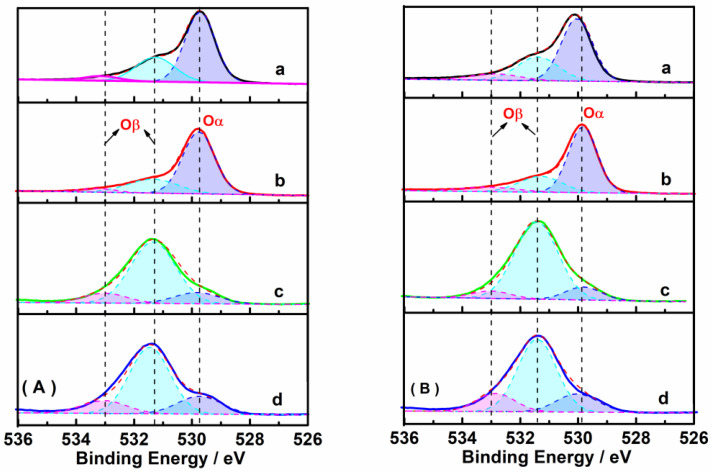
O 1s XPS spectra of the following samples (**A**) without and (**B**) with CO pretreatment: (a) Fe_2_O_3_-H, (b) Fe_2_O_3_-C, (c) 20% Fe_2_O_3_-H/13X, and (d) 20% Fe_2_O_3_-C/13X. Blue peak area represented lattice oxygen O_α_ species, while green and pink peak area, chemisorbed oxygen O_β_ species.

**Figure 8 ijms-23-12134-f008:**
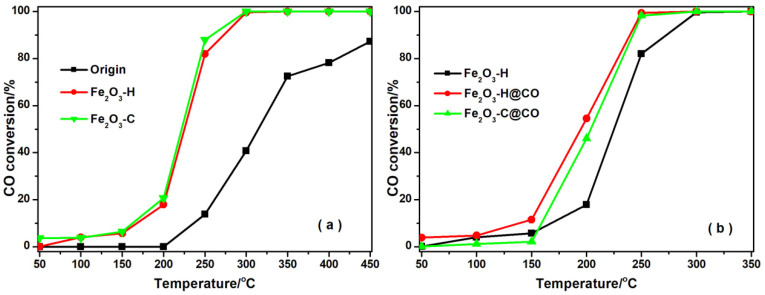
CO oxidation activities on (**a**) the original Fe resources, Fe_2_O_3_-H and Fe_2_O_3_-C; and (**b**) the Fe_2_O_3_-H, Fe_2_O_3_-H@CO and Fe_2_O_3_-C@CO.

**Figure 9 ijms-23-12134-f009:**
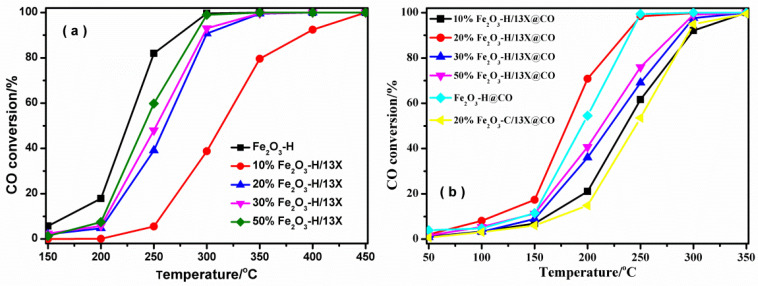
CO conversion as a function of reaction temperature on (**a**) the x% Fe_2_O_3_-H/13X; and (**b**) the x% Fe_2_O_3_-H/13X@CO and 20% Fe_2_O_3_-C/13X @CO.

**Figure 10 ijms-23-12134-f010:**
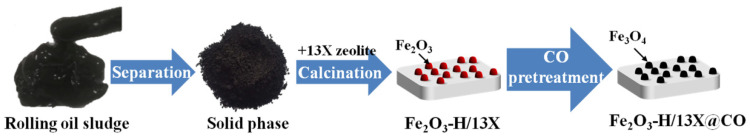
Schematic illustration of the Fe_2_O_3_-H/13X@CO preparation process.

**Table 1 ijms-23-12134-t001:** Peak-fitting quantitative results of Fe 2p and O 1s spectra of different samples.

Sample	Peak Location (eV)			Fe^2+^/(Fe^2+^ + Fe^3+^) ^a^	O_a_/O ^b^	O_b_/O ^c^
	Fe^2+^oct	Fe^3+^oct	Fe^3+^tet			
Fe_2_O_3_-H	-	-	-	-	66.4%	33.6%
Fe_2_O_3_-C	-	-	-	-	71.2%	28.8%
20% Fe_2_O_3_-H/13X	-	-	-	-	18.0%	82.0%
20% Fe_2_O_3_-C/13X	-	-	-	-	19.4%	80.6%
Fe_2_O_3_-H@CO	710.06	711.24	713.12	29.6%	61.2%	38.8%
Fe_2_O_3_-C@CO	709.92	711.15	712.92	27.1%	68.2%	31.8%
20% Fe_2_O_3_-H/13X@CO	709.90	711.30	713.06	34.6%	11.5%	88.5%
20% Fe_2_O_3_-C/13X@CO	709.9	711.15	712.98	20.1%	17.5%	82.5%

^a^ Fe^2+^/(Fe^2+^ + Fe^3+^) = S_Fe_^2+/^S_(Fe_^2+^ + _Fe_^3+^_)_. ^b^ O_a_/O = O_a_^/^S_(Oa+Ob)_. ^c^ O_b_/O = O_a_^/^S_(Oa+Ob)_.

**Table 2 ijms-23-12134-t002:** Summary of reported T100 on Fe based catalysts in CO oxidation.

NO	Catalysts	T100/°C	Ref.
1	Fe_2_O_3_/13X	250	This work
2	Ce-Fe	275	[23]
3	Fe_2_O_3_/Al_2_O_3_	300	[39]
4	Fe_2_O_3_/Al_2_O_3_	278	[40]
5	Fe_2_O_3_ rod	370	[41]
6	Fe_2_O_3_	288	[42]
7	Fe_2_O_3_/TiO_2_	260	[43]
8	Fe_2_O_3_/Al_2_O_3_	300	[44]
9	Fe_2_O_3_	350	[45]
10	Fe/Al-pillared bentonite	>400	[46]

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
