# Peer review of "Solid Fe Resources Separated from Rolling Oil Sludge for CO Oxidation"

_ijms, 2022, doi:10.3390/ijms232012134_

Round 1
Reviewer 1 Report
In this work, Gao et al. extracted solid Fe sources from ROS and then used them for CO oxidation after proper heat treatment. The novelty of the work is not discussed properly in the introduction section. Moreover, the manuscript requires sufficient improvement in English. Hence, I recommend it for the second round of revision. My specific comments are given below:
1) Fig.S2 is confusing. Please correct it.
2) Figures' qualities are very poor.
3) Line 14 in the abstract section; direct decomposition of ferric nitrate (Fe2O3-C) is confusing. Please correct.
4) Figure captions are not appropriate, especially Fig.2 and Fig.6. Please correctly write all the Figure captions so that readers should not get confused.
5) The XRD results of the Fe solid sources have not been explained properly. There was a clear phase transition from F2O3 to Fe3O4. Even the phase of Fe2O3 has not been discussed. Was it alpha or gamma? These papers might be helpful: https://iopscience.iop.org/article/10.1088/1361-6528/ac137a/meta; https://www.sciencedirect.com/science/article/abs/pii/S0167577X19313795
Author Response
In this work, Gao et al. extracted solid Fe sources from ROS and then used them for CO oxidation after proper heat treatment. The novelty of the work is not discussed properly in the introduction section. Moreover, the manuscript requires sufficient improvement in English. Hence, I recommend it for the second round of revision. My specific comments are given below:
Question 1. Fig.S2 is confusing. Please correct it.
.Answer: Thanks for reminding us. We have revised x-axis title from “Temperature” to “Reaction temperature” and also added “Calcined temperature” in the Fig.S2.
Question 2. Figures' qualities are very poor.
Answer: Thanks for the reviewer’s advice. We have improved all figures’ resolutions to 600 dpi.
Question 3. Line 14 in the abstract section; direct decomposition of ferric nitrate (Fe2O3-C) is confusing. Please correct.
Answer: Thanks for reminding us. We have revised “…prepared by direct decomposition of ferric nitrate (Fe2O3-C)” into “…prepared by the calcinations of ferric nitrate (Fe2O3-C)”. (Line 24–25)
Question 4. Figure captions are not appropriate, especially Fig.2 and Fig.6. Please correctly write all the Figure captions so that readers should not get confused.
Answer: Thanks for the reviewer’s advice. We have checked all the figure captions and revised inappropriate ones in the revised manuscript. (Line 570; Line 594)
Question 5. The XRD results of the Fe solid sources have not been explained properly. There was a clear phase transition from Fe2O3 to Fe3O4. Even the phase of Fe2O3 has not been discussed. Was it alpha or gamma? These papers might be helpful: https://iopscience.iop.org/article/10.1088/1361-6528/ac137a/meta; https://www.sciencedirect.com/science/article/abs/pii/S0167577X19313795
Answer: Thanks for the reviewer’s advice. I have discussed the Fe2O3 phase in the revised manuscript and referred the above two papers. By screening the XRD database, the peaks located at 24.1, 33.2, 35.6, 40.8 and 49.6o are assigned to these of a-Fe2O3 phase. (Line 124–125)
Reviewer 2 Report
Solid Fe Resources Separated from Rolling Oil Sludge for CO Oxidation
Wei Gao et al.
In the process of cool rolling of stainless steel, a rolling oil is used for refrigeration and lubrication. During exploitation, it becomes increasingly contaminated with iron particles, and hence needs to be discharged after long-term use as rolling oil sludge (ROS).
According to authors’ proposal an efficient recycling of valuable resources from ROS can be gained firstly by hydrogenation of it in order to separate an upgraded rolling oil and Fe particles, and secondly to use the latter, after calcination in air, as an active catalyst for CO oxidation. A wide spectrum of methods (XRD, TEM, XPS, TPD and O2-TPD) has been used for characterization of the above-mentioned catalyst and measurements of its activity have been performed. As a result, the state of the catalyst is well documented. As shown by the authors, the activity of the catalyst is greatly enhanced by milling it with 13X zeolite. The most active system, 20 wt%Fe2O3/13X showed 71% CO conversion at 200°C and 100% conversion at 250°C.
The idea is innovative, in line with the Principles of Green Chemistry, and in the opinion of the reviewer, is worth dissemination. Therefore, I recommend accepting the manuscript for publication after the following minor corrections have been made:
line 20 “…250 â—¦C” … should be replaced by “250°C” and such corrections should be made in the whole manuscript
line 62 Chemical names: “ethyl benzene”, “cyclo-hexane” written separately should be written together
line 295 … “6 Mpa” should be replaced by “6 MPa”
Author Response
In the process of cool rolling of stainless steel, a rolling oil is used for refrigeration and lubrication. During exploitation, it becomes increasingly contaminated with iron particles, and hence needs to be discharged after long-term use as rolling oil sludge (ROS).
According to authors’ proposal an efficient recycling of valuable resources from ROS can be gained firstly by hydrogenation of it in order to separate an upgraded rolling oil and Fe particles, and secondly to use the latter, after calcination in air, as an active catalyst for CO oxidation. A wide spectrum of methods (XRD, TEM, XPS, TPD and O2-TPD) has been used for characterization of the above-mentioned catalyst and measurements of its activity have been performed. As a result, the state of the catalyst is well documented. As shown by the authors, the activity of the catalyst is greatly enhanced by milling it with 13X zeolite. The most active system, 20 wt%Fe2O3/13X showed 71% CO conversion at 200°C and 100% conversion at 250°C.
The idea is innovative, in line with the Principles of Green Chemistry, and in the opinion of the reviewer, is worth dissemination. Therefore, I recommend accepting the manuscript for publication after the following minor corrections have been made:
Question 1. line 20 “…250 â—¦C” … should be replaced by “250°C” and such corrections should be made in the whole manuscript
Answer: Thanks for reminding us. We have replaced “â—¦C” into “â—¦C” in the revised manuscript.
Question 2. line 62 Chemical names: “ethyl benzene”, “cyclo-hexane” written separately should be written together
Answer: Thanks for reminding us. We have corrected the related words. (Line 96–97)
Question 3. line 295 … “6 Mpa” should be replaced by “6 MPa”
Answer: Thanks for reminding us. We have revised “6 Mpa” into “6 MPa”. (Line 316)
Reviewer 3 Report
The authors reported the recycling of solid Fe resources from ROS by a catalytic hydrogenation technique and its catalytic performances for CO oxidation. The solid Fe resources, after calcination in air (Fe2O3-H), exhibited comparable activity to these prepared by direct decomposition of ferric nitrate (Fe2O3-C), suggesting that the solid resources have excellent recycling value when used as raw materials for CO oxidation catalyst preparation. Further studies to improve the catalytic performances, by supporting the materials on high surface area 13X zeolite and by pretreating the materials with CO atmosphere, showed that the CO pretreatment greatly improved the CO oxidation activity and the best activity was achieved on the 20 wt.%Fe2O3-H/13X sample with complete CO conversion at 250 oC. The work is of some interesting. Therefore, I think it can be accepted for publication after minor revisions. My concerns are listed below.
1. The reaction mechanism of CO oxidation is suggested to be discussed in depth in combination with the experimental results. The related literatures can be referenced, such as THE JOURNAL OF CHEMICAL PHYSICS 134, 034305 (2011).
2. Please provide a scheme to show the process of the recycling of solid Fe resources from ROS.
3. It is informative for readers to add the Fe-based catalysts reported for comparison, such as Chinese Chemical Letters 31, 1201-1206 (2020).
Author Response
The authors reported the recycling of solid Fe resources from ROS by a catalytic hydrogenation technique and its catalytic performances for CO oxidation. The solid Fe resources, after calcination in air (Fe2O3-H), exhibited comparable activity to these prepared by direct decomposition of ferric nitrate (Fe2O3-C), suggesting that the solid resources have excellent recycling value when used as raw materials for CO oxidation catalyst preparation. Further studies to improve the catalytic performances, by supporting the materials on high surface area 13X zeolite and by pretreating the materials with CO atmosphere, showed that the CO pretreatment greatly improved the CO oxidation activity and the best activity was achieved on the 20 wt.%Fe2O3-H/13X sample with complete CO conversion at 250 oC. The work is of some interesting. Therefore, I think it can be accepted for publication after minor revisions. My concerns are listed below.
Question 1. The reaction mechanism of CO oxidation is suggested to be discussed in depth in combination with the experimental results. The related literatures can be referenced, such as THE JOURNAL OF CHEMICAL PHYSICS 134, 034305 (2011).
Answer: Thanks for the reviewer’s advice. Based on the above results, the catalytic mechanism of CO oxidation on the Fe2O3-H/13X@CO was then discussed. It is generally believed that iron oxide catalysts follow the redox mechanism in CO oxidation. For supported-Fe catalysts, CO molecular firstly adsorbs on the surface active sites, and is subsequently oxidized by lattice oxygen in iron oxides. Several medium CO32- or HCO3- species are formed, which are further converted into CO2. Correspondingly, reduced active sites in the former step are in turn oxidized to initial state by adsorbed O2 to become available for further reaction cycle. Combined with characterization and activity results, Fe2O3-H/13X@CO showed the best CO oxidation activity possibly due to the following reasons: Fe species with low valence states can generate more oxygen vacancies that speed up the reaction rate. Furthermore, 20% Fe2O3-H/13X@CO surface provided sufficient chemical chemisorbed oxygen species for reduced active sites’ regeneration and exhibited excellent reducibility for CO molecules’ oxidation, which also contributed to its best CO oxidation performance. (Line 289–302)
Question 2. Please provide a scheme to show the process of the recycling of solid Fe resources from ROS.
Answer: Thanks for the reviewer’s advice. We have added the scheme of the catalyst preparation process by utilizing solid Fe resources from ROS (see Fig. 10). (Line 335–336; Line 621–623)
Question 3. It is informative for readers to add the Fe-based catalysts reported for comparison, such as Chinese Chemical Letters 31, 1201-1206 (2020).
Answer: Thanks for the reviewer’s advice. We have introduced this work (Chinese Chemical Letters 31, 1201-1206 (2020)) in the introduction. Tian et al. synthesized Cu-Fe-Co ternary oxides thin film supported on copper grid mesh for CO oxidation, showing excellent catalytic activity owing to synergistic effects of chemisorbed oxygen species, electrical resistivity, easy mass transferability, etc. (Line 91–94)
Round 2
Reviewer 1 Report
The revised manuscript is now in better shape to be accepted for publication.